# Stomatal and Non-Stomatal Leaf Traits for Enhanced Water Use Efficiency in Rice

**DOI:** 10.3390/biology14070843

**Published:** 2025-07-10

**Authors:** Yvonne Fernando, Mark Adams, Markus Kuhlmann, Vito Butardo Jr

**Affiliations:** 1Department of Chemistry and Biotechnology, Swinburne University of Technology, Hawthorn 3122, Australia; ylfernando@swin.edu.au (Y.F.); maadams@swin.edu.au (M.A.); 2Institute of Plant Genetics and Crop Plant Research (IPK), 06466 Gatersleben, Germany; kuhlmann@ipk-gatersleben.de

**Keywords:** climate adaptation, crop sustainability, leaf physiology, rice breeding, water management

## Abstract

Growing rice requires a large amount of fresh water, but with increasing water shortages, improving the efficient use of water is essential. Many efforts have focused on the traits of stomata, tiny pores on leaves that control water loss and gas exchange, but other non-stomatal leaf traits, such as leaf structure, internal water movement, and biochemical processes, also play a crucial role. This review examines how both stomatal and non-stomatal leaf traits interact to influence water use efficiency (WUE) in rice. The findings suggest that optimising stomatal traits alone is not enough because non-stomatal factors significantly affect how efficiently leaves use water. By combining advanced technologies, such as high-throughput plant screening, genetic analysis, and computer modelling, researchers can identify the best mix of traits to develop rice varieties that use water more efficiently without reducing their yield. This integrated breeding approach can help create climate-resilient rice and improve the WUE in other cereal crops, supporting global food security in the face of water scarcity.

## 1. Introduction

Water scarcity threatens global food security, mainly because over half the world’s population depends on water-intensive crops like rice *(Oryza sativa* L.) [1,2]. There is substantial evidence that climate change is exacerbating water scarcity, and enhancing the water use efficiency (WUE) of rice crops has become a significant issue for sustaining agricultural productivity [3,4]. Consequently, in addition to testing alternative agronomic methods [5,6,7], researchers worldwide are seeking to breed rice varieties with greater WUE [8,9,10,11,12]. Plant leaves play a pivotal role in water exchange, acting as the primary site for carbon dioxide uptake and water vapour loss through transpiration [4]. Historically, efforts to improve WUE have focused predominantly on stomatal traits, such as stomatal density, size, and conductance. There is, however, mounting evidence that non-stomatal leaf traits, including photosynthetic efficiency, leaf anatomy, and biochemical composition, significantly influence WUE in rice [13,14,15].

Rice cultivation spans a diverse range of agroecosystems, from traditional continuously flooded paddies to water-limited upland and dryland environments. In flooded systems, WUE-related traits may enhance the efficient use of water resources and reduce water input requirements without imposing stress on the crop, whereas in upland systems, these traits are critical for survival, resilience, and yield stability under drought conditions. The physiological and anatomical mechanisms underlying WUE, such as stomatal behaviour, root architecture, and osmotic adjustment, therefore differ in their functional importance depending on the cultivation system. Recognising this spectrum is essential for developing rice varieties adapted to varying water regimes.

WUE can be broadly defined as the ratio of the biomass produced or the yield obtained to the amount of water used [9,12]. However, the concept of WUE encompasses multiple scales and definitions depending on the context of measurement. At the leaf level, WUE is typically expressed as the ratio of carbon assimilation to water loss through transpiration [10]. This can be quantified as intrinsic water use efficiency (WUEi), defined as the ratio of the net CO_2_ assimilation rate (An) to stomatal conductance (g_s_), or alternatively computed as the instantaneous water use efficiency (WUEinst), calculated by dividing An by the transpiration rate (E) [10,11,16]. These leaf-level measurements provide insights into the physiological basis of WUE and are influenced by stomatal and non-stomatal leaf traits. Measurement techniques for leaf-level WUE in rice include gas exchange analysis using infrared gas analysers, which allow for the simultaneous measurement of photosynthesis and transpiration rates [11,17] and carbon isotope discrimination (Δ^13^C) over the period of leaf development [18,19]. Additionally, chlorophyll fluorescence techniques can provide insights into photosynthetic efficiency, which is closely linked to WUE [20].

At the whole-plant level, WUE can be assessed as the ratio of biomass accumulation to cumulative water use [21,22,23,24]. This approach captures both productive and non-productive water losses. Field-scale measurements often focus on yield-based WUE, calculated as the crop yield per unit of water applied or lost through combined evaporation and transpiration [12]. Recent advances in high-throughput phenotyping, including thermal imaging and remote sensing techniques, enable more rapid and precise measurements of WUE-related traits in large populations [25]. These technologies, combined with genomic approaches, are accelerating the identification of genetic determinants of WUE in rice. Understanding the physiological mechanisms underlying WUE variability at the leaf and whole-plant level is essential for developing rice varieties with improved drought tolerance and reduced water requirements.

This review aims to critically evaluate the roles of stomatal and non-stomatal leaf traits in determining WUE in rice and explore their potential for improving water use through breeding. We also explore intricate interactions among these traits and their combined impact on water conservation and carbon assimilation. We examine the complex interplay of leaf traits to identify novel strategies for breeding rice varieties with a superior WUE without compromising their yield potential. Included in our consideration are (1) the varying scales of WUE measurement, from leaf-level gas exchange to whole-plant carbon isotope discrimination; (2) recent advances in the understanding of stomatal behaviour, mesophyll conductance, leaf anatomy, and metabolic adaptations to water stress; and (3) emerging technologies, such as high-throughput phenotyping and multi-omic approaches. By integrating these factors, we aim to provide a more holistic understanding of breeding programs for WUE in rice, including potential synergies and trade-offs. Finally, we consider how agronomic practices [26,27] and environmental factors interact with leaf traits to influence WUE under field conditions. This review was conducted through a systematic literature search of Web of Science, Scopus, and Google Scholar, covering studies from 1984 to 2025 focused on stomatal and non-stomatal leaf traits related to WUE in rice. Priority was given to peer-reviewed research encompassing both field and controlled-environment studies, particularly those addressing genetic, physiological, and anatomical analyses relevant to breeding for improved WUE.

## 2. Stomatal Leaf Traits and WUE in Rice

Stomatal leaf traits refer to the characteristics and properties of the stomata, including their density, size, distribution, and conductance, which are regulated by controlling the gas exchange between leaves and the atmosphere. As highlighted in a recent herbarium study of plant adaptation to climate change [28], there is extensive knowledge of the dozens of genes involved in stomatal development (in terms of density, size, and distribution).

### 2.1. Stomatal Density, Size, and Arrangement

Recent studies have demonstrated that modifying stomatal density and size through crop engineering offers a promising approach to enhancing WUE and mitigating water scarcity challenges in agricultural crops [4,29]. Rice exhibits unique stomatal structures compared to other plant species, with greater stomatal densities and smaller stomatal sizes. The stomatal complex in rice comprises two dumbbell-shaped guard cells surrounded by subsidiary cells that regulate the stomatal aperture [30,31].

Four distinct models of leaf stomata were described in mutant populations of the purple rice cultivar, Jao Hom Nin, which varied in stomatal density and size [3]. Rice varieties with a low stomatal density and a smaller stomatal size exhibited improved WUE and biomass production under both field and greenhouse conditions. Similarly, three purple rice mutants were adapted to severe water stress conditions during the reproductive stage of growth, which resulted in a low stomatal density and greater WUE under restricted water conditions [32]. Despite these findings, gaps remain regarding the physiological trade-offs of reduced stomatal density, particularly under fluctuating environmental conditions. While smaller, fewer stomata conserve water and they may limit CO_2_ uptake under high demand. These dynamics are not yet fully quantified across diverse environments and growth stages.

The genetic manipulation of stomatal traits has shown promise in improving WUE. The high-yielding rice variety IR64 was engineered to reduce stomatal density by enhancing the expression of *OsEPF1*, a gene involved in rice epidermal patterning [33]. The resulting plants exhibited reduced stomatal conductance and enhanced water conservation. These traits were interpreted as having improved drought tolerance while maintaining or improving yields under elevated atmospheric CO_2_ and high-temperature conditions. A previous study explored the impact of the arrangement of leaf stomata on the physiological responses of rice to differing levels of water availability [34]. Yet, the implications of this spatial arrangement for canopy-level water and carbon fluxes remain underexplored. They proposed that a greater stomatal density in the upper leaves (the flag leaf and second leaf) with a reduced density but larger stomata in the lower leaves (third and fourth leaves) constituted an adaptive strategy (reduced water loss) for genotypes under conditions of limited water availability. The study concluded that this stomatal configuration could enhance water productivity in rice under aerobic conditions without a reduction in yield.

While much of the current understanding of stomatal traits in rice comes from controlled environments, field-based evaluations reveal important cultivar-specific responses and environmental interactions influencing WUE under real-world drought conditions. Field measurements across diverse genotypes have shown considerable variability in stomatal conductance (g_s_) (discussed further in Section 2.2), density, and size. For example, a study evaluating 64 rice accessions and high-yielding cultivars found that although the stomatal density did not directly correlate with g_s_, modern cultivars typically exhibited a higher density and moderate specific conductance, whereas traditional varieties had a lower density but larger stomata. These findings underscore the complex interactions among stomatal traits and highlight their potential for optimisation through breeding to improve photosynthesis and WUE in the field [35]. Complementing this, a multi-location field study demonstrated that drought-tolerant breeding lines, including Sahbhagi Dhan, consistently had longer flag leaves and a lower stomatal density compared to the drought-susceptible cultivar IR64. These traits were positively linked to the grain yield and harvest index under drought across India, Bangladesh, Nepal, and the Philippines, emphasizing the importance of validating stomatal traits under field conditions to guide breeding strategies for drought resilience and sustainable productivity [36]. Overall, field studies show that rice cultivars employ distinct stomatal strategies to manage water scarcity, with drought-tolerant varieties generally exhibiting a lower stomatal density and adaptive leaf traits that promote yield stability. These traits interact with complex environmental factors absent in controlled settings, underscoring the necessity of validating laboratory findings under real field conditions to ensure effective breeding.

### 2.2. Stomatal Conductance and Aperture

Leaf stomatal conductance (g_s_) is a critical physiological parameter influenced by a wide range of environmental and plant factors, including light, temperature, humidity, CO_2_ concentration, and plant water status [4,37]. The relationship between g_s_ and WUE involves a trade-off between maximising CO_2_ uptake for photosynthesis and minimising water loss through transpiration. Recent studies have focused on understanding genetic and molecular mechanisms that regulate stomatal conductance and aperture in rice. A previous study revealed that the precise control of the stomatal aperture is crucial to maintaining an optimal balance between photosynthesis and water conservation [38]. The study identified genes such as *EPF2*, *OSA1*, and *SLAC1* involved in stomatal regulation, enabling the enhancement of WUE through genetic manipulation. However, the long-term physiological impacts of manipulating these genes under fluctuating climate conditions remain poorly understood. In particular, the extent to which these molecular interventions affect gas exchange efficiency under combined heat and water stress is underexplored. Comparative evidence from model species like *Arabidopsis* suggests variation in stomatal responsiveness and downstream signaling efficiency, indicating that gene function may be context-dependent across crops [39].

### 2.3. Regulation of Transpiration and Carbon Dioxide Uptake

Abscisic acid (ABA) is widely believed to have a central role as a signal for the closure of stomata during water stress [37,40]. Other hormones, such as cytokinins and gibberellins, also contribute to stomatal regulation. Plant circadian rhythms reflect the significance of stomatal conductance, ensuring that stomatal opening and closing are synchronised with environmental light and temperature patterns [40]. This temporal coordination helps optimise WUE and CO_2_ uptake and contributes to plant growth and survival under varying environmental conditions. Similarly, ion transport and osmotic adjustments within stomatal guard cells facilitate rapid changes in the stomatal aperture, balancing the need for CO_2_ uptake with water conservation. Fine-tuning these processes enhances plant survival in diverse and changing environments [3,40]. The movement of ions such as potassium (K^+^), chloride (Cl^−^), and calcium (Ca^2+^), along with malate (C_4_H_4_O_4_^2−^), causes water to enter guard cells, leading to the stomata opening for CO_2_ uptake during photosynthesis. Conversely, water follows when ions efflux from the guard cells, causing the stomata to close, thereby reducing water loss (Figure 1A). The fine-tuning of this mechanism is essential for balancing the demand for CO_2_ with the need for water conservation, especially under drought stress [41,42]. Despite this mechanistic clarity, limited studies have explored how these responses are modulated across rice genotypes with varying levels of limited water stress or drought resilience. Moreover, cross-species comparisons of hormonal regulation and ion channel dynamics remain scarce, leaving uncertainties about whether similar mechanisms operate with equal efficiency in monocot cereals versus dicot plants.

In addition to stomatal and cuticular transpiration, grasses such as rice also release water through hydathodes: specialised structures located at leaf margins that facilitate guttation under the conditions of a high amount of soil moisture and low transpiration demand. Hydathodes, unlike stomata, remain open and serve as low-resistance pathways for liquid exudation, playing a role in water release and nutrient exudation [43]. This process complements transpiration and may contribute to maintaining xylem flow when the evaporative demand is low. Furthermore, rice leaves contain well-defined vascular bundles composed of xylem and phloem, which are responsible for the transport of water, nutrients, and photosynthates. These vascular structures are closely associated with companion cells and are integral to leaf physiology, supporting both hydraulic conductance and photosynthate distribution [44].

## 3. Non-Stomatal Leaf Traits and WUE in Rice

Non-stomatal leaf traits in rice encompass the structural and biochemical characteristics of the leaf, which influence photosynthetic efficiency and WUE independent of stomatal behaviour.

### 3.1. The Role of ΦPSII in Photosynthetic Efficiency

In this review, photosynthetic efficiency is categorised as a non-stomatal leaf trait because it primarily reflects the biochemical and physiological processes within the leaf, including the efficiency of the photosynthetic machinery, such as chloroplasts, enzymes, and photosystems like ΦPSII in converting light energy into chemical energy. Photosynthetic efficiency refers to the ability of plants to convert light energy into chemical energy during photosynthesis [45]. Non-stomatal factors, such as the biochemical capacity of the mesophyll cells, chlorophyll content, and the efficiency of the photosynthetic machinery, play a pivotal role in determining photosynthetic efficiency [4,18,46,47]. These factors can enhance the plant’s ability to assimilate CO_2_ under varying environmental conditions, thereby optimising WUE. By improving photosynthetic efficiency, plants can achieve higher rates of CO_2_ assimilation per unit of water transpired. This balance is achieved by optimising stomatal conductance, so that stomata remain partially open to allow CO_2_ uptake while reducing excessive water loss. Improved mesophyll conductance, Rubisco efficiency, or enhanced photoprotection mechanisms can also contribute to a higher intrinsic WUE under drought stress [48]. This balance between CO_2_ uptake and water conservation is essential for plant productivity and resilience, making photosynthetic efficiency a key target for crop improvement strategies [49].

### 3.2. Leaf Anatomy

Mesophyll structure, vein density, and bulliform cells significantly impact CO_2_ diffusion, light distribution, and water management within leaves [50]. Mesophyll conductance (g_m_), an important factor influencing CO_2_ diffusion, is discussed in more detail in Section 3.5. Leaf thickness and cell arrangements influence light distribution and absorption, while chloroplast positioning optimises light harvesting and minimises photodamage. However, these anatomical features often involve trade-offs. For instance, it has been reported that an increased vein density in rice is correlated with greater photosynthetic rates but a reduced WUE [51], highlighting the complex interplay between structural traits and physiological efficiency.

Aquaporins, a family of integral membrane proteins, play a dual role in regulating both water transport and CO_2_ diffusion across leaf tissues, thereby influencing photosynthetic efficiency and WUE in rice. While CO_2_ is abundant in the atmosphere, its diffusion into the chloroplasts, where photosynthesis occurs, is limited by mesophyll conductance, which includes movement through intercellular air spaces, cell walls, and membranes. Certain aquaporins, particularly plasma membrane intrinsic proteins (PIPs), facilitate this process by enabling CO_2_ to cross cell membranes more efficiently [52]. In parallel, aquaporins influence leaf hydraulic conductance, which refers to the efficiency of water movement from the xylem to the sites of evaporation within the leaf. This conductance affects stomatal behaviour and transpiration, thereby indirectly modulating photosynthesis and WUE [50,53]. Environmental factors such as nitrogen availability can regulate aquaporin expression and activity, further linking nutrient status to water and carbon dynamics in leaves [48]. A balanced distribution of aquaporins in the mesophyll, bundle sheath, and vascular tissues is considered optimal for improving WUE and enhancing crop resilience under climate stress [54,55].

Studies have emphasised the significance of internal leaf anatomy, including aerenchyma and intercellular air spaces, in regulating CO_2_ diffusion to the sites of photosynthesis. CO_2_ must first dissolve in the thin water layer lining the mesophyll cell walls before diffusing into the symplast and reaching the chloroplasts. Therefore, a greater intercellular surface area, supported by well-developed aerenchyma, enhances both CO_2_ diffusion efficiency and evaporative capacity. This anatomical feature contributes to g_m_ and plays a critical role in photosynthetic performance and WUE, particularly under stress conditions (Figure 1B) [56,57,58].

Studies have further demonstrated that a midvein-deficient mutant shows a higher vein density and photosynthetic efficiency but a significantly lower WUE. Although the study did not include water stress conditions, the mutants are still recommended as promising rice breeding material due to their enhanced photosynthetic efficiency [51]. In a glasshouse experiment with limited water (60% field capacity), it has been observed that high-leaf-mass-area (LMA) mutants exhibited a higher WUE and a more significant total number of veins than low-LMA mutants, which had a lower WUE. However, high-LMA mutants also showed a lower vein density (veins per centimetre) and fewer bulliform cells than their low-LMA counterparts [14].

Abaxial stomatal dominance, where stomata are mainly located on the leaf’s lower surface, helps reduce water loss by limiting stomatal exposure to direct sunlight and wind. This spatial trait can enhance WUE in rice, especially under drought or high vapor pressure conditions. When combined with other anatomical traits like a reduced stomatal density and thicker cuticles (as discussed in Section 3.3), abaxial dominance offers a promising target for breeding rice varieties with improved drought resilience and efficient gas exchange [33,59].

### 3.3. Leaf Cuticles and Epicuticular Wax

Leaf cuticles and epicuticular wax (EW) are critical components in regulating non-stomatal water loss and enhancing WUE in rice. In rice, the β-Ketoacyl-CoA synthase (KCS) enzyme is essential for the elongation of very long-chain fatty acids, which are major components of cuticular wax [60]. It has been reported that an increased expression of the *OsCUT1* gene in rice resulted in improved drought tolerance and wax synthesis without reducing yield [61]. On the other hand, the *OsGL1-1* gene plays a crucial role in cuticular wax deposition and drought resistance in rice. Mutants with reduced *OsGL1-1* expression showed decreased cuticular wax deposition, increased water loss, and enhanced sensitivity to drought compared to wild-type plants [62]. Furthermore, previous research has demonstrated that EW traits contribute to rice resistance against pests, evidenced by EW rice mutants exhibiting reduced resistance to the rice water weevil (*Coleoptera: Curculionidae*) and the fall armyworm (*Lepidoptera*: *Noctuidae*) (Figure 1C) [63].

### 3.4. Metabolomic Changes in Leaves

Drought stress triggers complex metabolic responses in plants, regulated by gene activity, which impact vital processes like photosynthesis and respiration and assimilate translocation [64]. These changes help the plant adjust osmotically, manage reactive oxygen species (ROS), and stabilise cell membranes, which are crucial for maintaining function during drought.

Drought-tolerant rice cultivars often accumulate greater concentrations of osmoprotectants, including low-molecular-weight osmolytes such as glycine betaine and proline, as well as organic acids and soluble sugars [64,65]. These compounds help sustain cellular functions and contribute to defence mechanisms under water-limited conditions. Additionally, increased concentrations of antioxidants, phenolic compounds, and flavonoids have been observed in drought-stressed rice leaves, with flavonoids like kaempferol and quercetin playing a significant role in enhancing drought tolerance and modulating WUE [66].

### 3.5. Mesophyll Conductance (g_m_) and Intrinsic WUE

g_m_ is a key physiological parameter that controls the rates of CO_2_ assimilation [46,67]. Von Caemmerer and Evans (2010) found that increasing photosynthesis without altering stomatal function can improve WUE and that enhancing g_s_ could boost photosynthesis if water is not limited [68]. g_m_ is measured as the rate of CO_2_ diffusion from intercellular spaces to chloroplasts, which substantially affects WUE. Varieties with greater g_m_ also show improved carbon assimilation per unit of water transpired [15]. Additionally, they mentioned that manipulating g_m_ could increase CO_2_ availability at Rubisco, the key enzyme in photosynthesis that catalyses the fixation of CO_2_ into organic molecules in the Calvin cycle, potentially improving photosynthesis and WUE [68]. Furthermore, Flexas et al. (2013) stated that to simultaneously improve photosynthesis and WUE, genetic manipulations of g_m_ should be performed without parallel changes in g_s_ [69]. It was suggested that the optimal trait for enhancing WUE is an increased ratio of g_m_ to g_s_ [69].

For rice, g_m_ plays a significant role in transpiration efficiency (TE) and could affect WUE. Upland rice and *O. glaberrima* showed a stronger response of TE to the g_m_/g_s_ ratio under drought conditions, suggesting a link between g_m_ and WUE. Furthermore, it suggested that increasing rice’s g_m_/g_s_ ratio, potentially by modifying leaf anatomy, could enhance both TE and WUE under water-limited conditions [70].

Strategies like genetic manipulation or selective breeding can significantly improve WUE without compromising photosynthetic capacity. The α subunit of a heterotrimeric G protein regulates g_m_ and drought tolerance in rice, suggesting that modifying G protein signalling could enhance WUE in rice cultivars [71].

### 3.6. Leaf Canopy Architecture

Rice varieties with a higher total leaf area and optimised canopy structure exhibited improved WUE under water-limited conditions [34]. Rice mutants with LMA showed an increased WUE, increased photosynthetic capacity, and superior carboxylation efficiency compared to low-LMA mutants under both well-watered and water-limited conditions [14,72].

A glasshouse study conducted on the Azucena (tropical japonica) and IR64 (India) rice varieties revealed a correlation between increased leaf width and lower Δ^13^C. Leaves with higher widths have excellent leaf boundary layers that resist gaseous diffusion, which would lower the Δ^13^C and increase the WUE [73].

A higher crop density creates dense canopies and increases the thickness of the boundary layer, the layer of still air that surrounds each leaf. This reduces evaporative demand and water loss and potentially improves WUE. However, it may also limit CO_2_ diffusion, affecting photosynthesis [74,75].

Erect leaves, which form smaller angles with the stem, allow for greater light penetration to lower canopy layers, thereby enhancing the overall photosynthetic efficiency and reducing excessive transpiration in the upper canopy. This spatial leaf arrangement has been demonstrated in rice genotypes to improve light interception and optimise water use, contributing to a higher WUE [76,77]. Furthermore, studies on spatial leaf arrangement and canopy architecture in rice show that coordinated leaf positioning around the stem improves the light distribution throughout the canopy, enhances carbon assimilation, and reduces water loss. This trait is particularly beneficial in high-density planting systems, in which maximizing light use efficiency and conserving water are critical for sustaining productivity under variable environmental conditions [78,79].

### 3.7. Leaf Pubescence and Boundary Layer Resistance

Leaf hairiness (pubescence) has been understudied in rice despite its potential to enhance water conservation and photosynthetic efficiency. Rice varieties show differences in leaf hairiness, from smooth to densely hairy surfaces. These hairs form a boundary layer that slows water vapour diffusion, reducing transpiration and enhancing light reflectance, which can protect the leaf from excessive radiation (Figure 1D) [75,80].

The genetic feasibility of transferring this trait between rice varieties and the physiological benefits of leaf hairiness in terms of WUE in rice have been studied by introducing a gene from chromosome 6 from hairy-leafed *Oryza nivara* into the less hairy IR24 variety, creating an introgression line with the hairy leaf trait (IL-hairy). The IL-hairy plants exhibited warmer leaf surfaces under sunlight due to the increased boundary layer resistance conferred by the leaf hairs and lower transpiration rates under moderate and high light intensities, leading to a significantly higher photosynthetic WUE [75].

### 3.8. Carbon Fixation Efficiency

Carbon fixation efficiency is discussed in this review as a non-stomatal trait because it primarily reflects the internal biochemical capacity of the plant to assimilate carbon, beyond the regulation of gas exchange at the stomatal level. While stomatal conductance indeed influences CO_2_ availability in the intercellular spaces, the efficiency with which CO_2_ is fixed into carbohydrates, governed by enzymes like Rubisco, chloroplast function, and mesophyll conductance, is largely non-stomatal in nature.

Δ^13^C and the carbon isotope ratio (δ^13^C) are widely used indicators of WUE in C_3_ plants like rice. Δ^13^C reflects the degree of discrimination against ^13^CO_2_ during photosynthesis, which is influenced both by stomatal behaviour and the biochemical processes involved in carbon fixation [18,81]. Higher Δ^13^C values indicate greater discrimination against ^13^C, typically associated with higher stomatal conductance and a lower WUE [82]. Conversely, δ^13^C, which is derived from plant tissue and reflects the integrated effects of carbon assimilation over time, is negatively correlated with Δ^13^C [83]. A higher δ^13^C value suggests reduced stomatal conductance and an improved carbon fixation efficiency per unit water lost, thus serving as a reliable proxy for assessing WUE in rice, especially under water-limited conditions [84].

## 4. Integrating Stomatal and Non-Stomatal Traits for WUE Improvement

Improving WUE in rice requires an integrated approach that considers both stomatal and non-stomatal leaf traits. This will enable the development of more effective strategies for sustainable rice production under water-limited conditions.

### 4.1. Interactions and Trade-Offs Among Leaf Traits

Reducing stomatal density can enhance water conservation [33], but it may also limit CO_2_ uptake, potentially impacting photosynthetic efficiency. Conversely, increasing g_m_ can improve photosynthetic rates but may lead to greater water loss if not balanced with appropriate stomatal control [47]. Hence, some studies have highlighted the importance of considering multiple traits simultaneously. It has been demonstrated that rice varieties with both optimised stomatal traits and enhanced g_m_ exhibit superior WUE compared to those improved for either trait alone [85]. Further, it has been found that stomatal traits modulated the relationship between leaf nitrogen content and WUE, emphasizing the need for integrated trait selection in breeding programs [86].

### 4.2. Breeding Strategies for Optimising WUE Through Leaf Traits

Optimising WUE in rice through breeding involves selecting specific leaf traits that improve water management and photosynthetic performance. This multi-trait approach combines traditional breeding with advanced genetic and genomic tools, focusing on stomatal behaviour, leaf morphology, physiological traits, photosynthetic efficiency, and biochemical pathways to develop more water-efficient, productive rice varieties.

Advances in quantitative trait loci (QTL) mapping have allowed the identification of genetic regions linked to desirable leaf traits, such as leaf area, thickness, and orientation. By correlating genetic variations with these traits, QTL mapping helps breeders efficiently integrate these traits into new plant varieties [87,88,89].

Marker-assisted selection (MAS) allows for the rapid integration of WUE-related traits by using molecular markers associated with key traits like stomatal conductance, leaf morphology, and Rubisco efficiency. Genomic selection takes this a step further by utilising genome-wide data to predict and select for high WUE traits, streamlining the breeding process and reducing the time needed to develop new varieties. This approach leverages high-throughput genotyping and phenotyping to build predictive models that can accurately identify the best candidates for breeding programs focused on improving WUE [90,91,92].

Recent studies have highlighted the importance of integrating both stomatal and non-stomatal traits to enhance WUE in rice. Stomatal morphology and physiology operate in a coordinated manner, influencing both transpiration efficiency and carbon assimilation [49]. Environmental drivers shape stomatal traits, which in turn interact with broader leaf structural features to modulate WUE at the ecosystem level [93]. Complementing these findings, another study [13] has shown that rice genotypes with low-frequency, large-size stomata exhibited a higher intrinsic WUE and more responsive gas exchange under varying vapour pressure deficits. This study also linked epidermal patterning and the expression of key regulatory genes (e.g., *ERECTA*, *TMM*, and *YODA*) to stomatal development, reinforcing the need to integrate leaf surface traits with internal physiological and anatomical features. Collectively, these findings support a holistic breeding approach that considers the interplay of multiple leaf traits to develop rice cultivars resilient to climate variability and water-limited environments.

Understanding the heritability of stomatal and non-stomatal leaf traits is critical for effective breeding strategies to improve WUE in rice. Heritability measures the proportion of total phenotypic variation in a trait attributable to genetic variation among individuals within a population. It quantifies a trait’s potential to respond to selection and is a key parameter in plant breeding programs [94]. Key leaf anatomical traits, such as mesophyll porosity and vein density, have been reported to exhibit high broad-sense heritability values ranging from 0.743 to 0.996, based on multi-model genome-wide association studies (GWAS) in diverse rice accessions [95]. Similarly, a high level of heritability was observed for vein density, leaf width, and photosynthetic rate in 99 rice genotypes evaluated under field conditions, indicating strong genetic control and the potential for effective selection in breeding programs targeting WUE and yield improvement [96]. Advances in high-throughput phenotyping have enabled the identification of heritable image-based drought resistance traits (i-traits), with GWAS revealing 443 loci strongly associated with these traits, many co-localising with known drought resistance QTLs [97]. This demonstrates the feasibility of dissecting complex drought-related traits into simpler, heritable components for breeding applications. Furthermore, photosynthetic traits such as the maximum CO_2_ assimilation, carboxylation rate, electron transport rate, and triose phosphate utilisation (TPU) exhibit moderate to high broad-sense heritability (H^2^ = 0.63–0.73), indicating strong genetic control and selection potential for improving photosynthetic efficiency in rice [98]. Genetic advances of up to 17.7% suggest significant gains can be achieved through selection in breeding programs.

### 4.3. Agronomic Practices and Environmental Factors Influencing WUE

Anthropogenic water management is an important factor influencing plant performance. Inefficient irrigation methods lead to a low WUE and environmental issues. Several alternative rice production methods, such as aerobic rice, direct-seeded rice (DSR), alternate wetting and drying (AWD), and smart irrigation technologies, can improve WUE, though each has specific challenges like high costs or labour intensity [6,99]. Delayed permanent water (DPW) is an alternative water management practice extensively used in Australian rice cultivation to increase water productivity for drill-sown rice. In this method, the rice crop is sown and initially managed like conventional drill-sown rice, but the application of permanent water is delayed until at least 50 days after the first incidence of flush irrigation [100].

Further, it has been emphasised that to promote water-saving irrigation technology, countries should develop supportive policies, raise farmers’ awareness, and offer training and financial assistance tailoring water-saving technologies to specific regions, which can reduce irrigation water use and improve rice WUE, helping to address global water scarcity challenges [101]. As a country with a well-managed rice industry, the Australian government’s Climate-Smart Agriculture Program provides funding for innovative irrigation projects [102]. Educational campaigns like Brisbane’s “Every Drop Counts” raise awareness among farmers about water-saving practices [103]. Organizations such as Irrigation Australia offer training programs and financial subsidies to help farmers adopt new technologies [104].

Nutrient management such as applying balanced fertilisers, particularly nitrogen, in appropriate amounts can enhance plant growth and photosynthetic efficiency, leading to a better WUE [9]. Sowing techniques, such as direct seeding and having a proper transplanting density, can optimise plant establishment and water utilisation. Direct seeding generally achieves a higher WUE than traditional transplanting, though wet seeding by broadcast, a common method, results in a lower WUE compared to traditional transplanting [105]. Moreover, uncontrolled weeds compete with rice for water and nutrients. It has been found that while weed control does not affect total water consumption, it influences WUE through its impact on yield [106]. Higher temperatures can raise transpiration rates, but practices like mulching help regulate plant temperature and reduce water loss. Non-flooded plastic film mulching and wheat straw mulching are effective options for improving WUE in rice cultivation [7]. Therefore, optimising WUE in rice production requires multiple approaches, integrating breeding for better leaf traits, effective agronomic practices, and the consideration of environmental factors to achieve sustainable rice production in water-limited conditions.

### 4.4. WUE Optimization Strategies Between Traditional and Dryland Rice Farming Systems

Rice cultivation spans a diverse range of agroecosystems, from continuously flooded lowland paddies to water-limited upland or dryland fields. These contrasting environments impose distinct physiological constraints and selection pressures on rice plants, resulting in divergent strategies for optimising WUE. Figure 2 compares key physiological, anatomical, and agronomic traits of rice grown in traditional paddy systems versus dryland/upland systems.

In traditional irrigated paddy systems, the presence of standing water creates a saturated microenvironment with a high level of humidity and low evaporative demand. This decouples plant water status from atmospheric vapor pressure deficit (VPD), reducing transpirational losses [107]. Consequently, irrigated rice typically develops shallow root systems and extensive aerenchyma tissue to facilitate internal oxygen transport under anaerobic conditions. Stomatal conductance in these systems tends to be relatively high and less responsive to VPD fluctuations. Additionally, non-stomatal adaptations such as epicuticular wax deposition and leaf rolling are minimal, as the consistently moist conditions reduce the need for such drought avoidance traits [6,108,109,110].

In contrast, dryland or upland rice is cultivated under aerobic conditions in which water availability is often intermittent and limiting. These environments favour more conservative water use strategies. Upland rice varieties typically exhibit deeper and more highly branched root systems to access subsoil moisture. They also demonstrate tighter stomatal regulation, with greater sensitivity to both soil water potential and atmospheric VPD. Non-stomatal adaptations are more pronounced, including increased epicuticular wax accumulation, enhanced leaf rolling during drought stress, and the production of osmoprotectants such as proline and soluble sugars that help maintain cellular integrity under dehydration [6].

Management practices to enhance WUE also differ significantly between the two systems. In irrigated paddy fields, techniques like AWD have been adopted to reduce water inputs while maintaining yield stability. This method intermittently drains fields to encourage deeper rooting and reduce methane emissions. In dryland systems, practices such as mulching, deficit irrigation, and optimised planting densities are employed to reduce surface evaporation, improve soil moisture retention, and enhance root–soil contact [111,112]. These contrasting strategies underscore the importance of context-specific breeding and agronomic interventions. While some core physiological principles of WUE, such as efficient stomatal control and root system architecture, are relevant across systems, their functional significance and genetic regulation vary with environmental and management conditions.

## 5. Future Perspectives and Research Directions

The exploration of WUE in rice is entering an exciting new era, driven by technological advances and interdisciplinary approaches. This last section examines emerging technologies that are revolutionising our ability to study leaf traits and WUE, from high-resolution imaging techniques to multi-omics approaches It also addresses the key challenges facing researchers and breeders in developing water-efficient rice cultivars and discusses the broader implications of this work for sustainable agriculture and climate resilience. By highlighting these future directions, we aim to provide a roadmap for researchers and emphasise the critical importance of continued innovation in this field.

### 5.1. Emerging Technologies and Approaches for Studying Leaf Traits and WUE

Recent technological advancements have revolutionised our ability to study leaf traits and WUE in rice at unprecedented levels of detail and scale. These innovations span a wide range of disciplines, from advanced imaging techniques to molecular biology approaches, offering researchers powerful new tools to unravel the complexities of plant–water interactions. Confocal microscopy offers high-resolution, 3D images of leaf tissues, focusing on cellular structures and stomatal dynamics [113]. Its advantage lies in cellular-level visualisation, but it is limited to small sample sizes and controlled environments. Thermal imaging with infrared thermography measures canopy temperature, indicating plant water status and transpiration rates. This method is rapid and non-invasive, yet sensitive to environmental fluctuations such as wind and ambient temperature [114,115]. X-ray micro-computed tomography (Micro-CT) and magnetic resonance imaging (MRI) provide non-destructive 3D imaging of leaf anatomy and water distribution. These offer exceptional anatomical insights but require costly equipment and may not be feasible for field-scale studies [116,117]. Chlorophyll fluorescence imaging, such as pulse amplitude modulation (PAM) fluorometry, assesses photosynthetic efficiency. While informative, its interpretation requires specialised knowledge, and results can vary depending on leaf age and environmental stress [118,119]. Field-based platforms use sensors and imaging systems on tractors [120], while greenhouses and growth chambers utilise automated systems to monitor conditions [121]. Unmanned aerial vehicles (UAVs) equipped with multispectral, hyperspectral, and thermal cameras capture data on plant health and water status from above [122]. UAVs enable the rapid coverage of large areas but depend on the weather and require data-processing expertise [123]. Hyperspectral imaging captures detailed spectral information to assess physiological and biochemical traits [124]. Light detection and ranging (LIDAR) creates high-resolution 3D maps of plant structures, helping to evaluate leaf area and canopy architecture, as well as how these factors influence light interception, transpiration, and overall water use [125]. Both tools provide deep insights into spatial traits, though data complexity and processing time remain significant challenges.

In addition to the stable carbon isotopes in Δ^13^C, oxygen and hydrogen isotopes (δ18O and δ2H) offer insights into water sources and usage dynamics [126]. Near-infrared spectroscopy (NIRS) can also analyse the chemical composition of plant tissues, which correlates with water content and WUE [127]. Raman spectroscopy analyses molecular compositions and structures to study biochemical changes in leaves [128]. These spectroscopic methods are fast and non-destructive but may require calibration against destructive sampling methods [129].

Genomic approaches like CRISPR-Cas9 and gene editing, along with transcriptomics, proteomics, and metabolomics, are powerful tools for studying WUE. CRISPR-Cas9 enables precise gene modifications to improve WUE, while transcriptomics and proteomics analyse gene expression and protein profiles to identify key WUE regulators [130]. Metabolomics examines the complete set of plant metabolites to understand metabolic pathways related to water use and stress responses [131]. These molecular approaches allow for targeted breeding and trait discovery, though they are resource-intensive and require validation under field conditions.

Big data and machine learning are transforming the study of WUE by using predictive modelling to combine environmental, genomic, and phenotypic data for optimising breeding strategies. Machine learning algorithms analyse large datasets to identify patterns affecting WUE, such as soil moisture and climate conditions [132]. Satellite imagery and geographic information systems (GIS) provide large-scale vegetation, soil, and climate data to monitor and manage WUE across regions. Smart irrigation systems utilise soil moisture sensors and weather forecasts to optimise water use and reduce waste [133]. While these digital tools offer scalability and precision, their effectiveness depends on data quality, infrastructure, and model accuracy.

Compared to traditional methods, such as manual trait measurements, gas exchange analysis using portable photosynthesis systems, or basic soil moisture probes, emerging technologies offer greater speed, scale, precision, and data integration. However, they also demand greater technical expertise, infrastructure, and investments. A balanced application that combines traditional and emerging methods may provide the most comprehensive insight into rice physiology and WUE improvement [134].

### 5.2. Challenges in Developing Water-Efficient Rice Cultivars

Despite significant advances in our understanding of WUE in rice, the development of water-efficient cultivars faces several complex challenges. These obstacles span genetic, physiological, and agronomic domains, necessitating a multidisciplinary approach to breeding and crop improvement. Key challenges include the trade-off between improving WUE and potential reductions in biomass or yield. Genetic complexity is another challenge, as WUE is influenced by many genes and their interactions with the environment [73,135]. Additionally, the diversity of rice-growing ecosystems, from flooded paddies to rainfed fields, necessitates a multifaceted approach to breeding [136]. Therefore, variations in soil type, fertility, and water-holding capacity further complicate efforts to develop effective cultivars.

High-throughput phenotyping for WUE can be challenging due to the technical difficulty and cost of advanced technologies, which may not be accessible to all breeding programs, especially in developing countries. Additionally, traditional rice farming practices and the historical significance of rice cultivation may lead to resistance among traditional farmers to adopt new cultivars due to concerns about their performance or a preference for established methods. 

To address these challenges, multi-omics approaches offer promising tools for dissecting the complex genetic and physiological traits underlying WUE. A summary of these omics-based strategies and their applications in rice WUE research is provided in Table 1.

The implications for sustainable agriculture and climate resilience are as follows. The development of water-efficient rice cultivars has far-reaching implications that extend beyond immediate agricultural productivity, touching on global issues of food security, environmental sustainability, and climate change adaptation. By addressing these critical challenges, advances in rice WUE have the potential to transform agricultural practices and contribute significantly to sustainable development goals. Developing and using water-efficient rice cultivars can reduce irrigation needs, conserving water resources and supporting agriculture in water-scarce areas. This approach lowers energy requirements for water distribution, promotes sustainable farming, and ensures stable yields despite variable water availability, enhancing food security and farmer incomes, especially in drought-prone regions. By improving crop resilience to water stress, these cultivars help farmers adapt to climate change, minimize crop failure risks, and reduce economic losses. Additionally, efficient water use can decrease methane emissions from flooded fields and benefit local ecosystems by lessening the depletion of natural water bodies [1,146,147].

A comprehensive understanding of the interplay between stomatal and non-stomatal traits is vital for enhancing WUE in rice under climate variability. Stomatal traits directly influence transpiration and gas exchange, while non-stomatal traits, such as leaf morphology, cuticular wax load, and root architecture, contribute to water retention and drought adaptation. Together, these traits modulate the plant’s physiological response to biotic and abiotic stresses, offering complementary pathways to improve WUE without compromising yield potential. Integrating both trait types in breeding strategies supports the development of cultivars that are better adapted to diverse and water-limited environments, aligning with the broader goals of sustainable agriculture and climate resilience [13,93].

### 5.3. Interaction with Root Traits

Another important consideration that is often overlooked is that the WUE of rice involves a complex interaction between leaf and root traits. Deeper rooting in rice genotypes increases WUE under drought conditions by reducing the internal concentration of CO_2_ (Ci) within the leaf, highlighting variability in root traits and gas exchange parameters among genotypes. Extensive root systems help plants access more water, but efficient stomatal control is vital to prevent excessive water loss. Roots signal leaves to adjust stomatal conductance based on soil water availability. Good WUE enhances root growth by providing more photosynthates, further improving water acquisition [11]. It was found that the gas and transpiration rates decrease during soil drying before the leaf water potential declines, indicating the importance of root signaling [148]. Hence, the interaction of leaf and root traits in enhancing WUE is also an underexplored area of study that warrants further investigation. 

To support future research and trait-based breeding strategies, Appendix A provides a comprehensive list of rice genes associated with both stomatal and non-stomatal leaf traits affecting WUE. The table includes gene functions, locus identifiers (IDs), and directional associations with WUE, offering a valuable resource for researchers aiming to link physiological traits with genetic targets. Locus IDs correspond to the Rice Annotation Project and were verified using the SNP-Seek database from the International Rice Research Institute [149]. Genes are organised into two main categories: (A) stomatal traits [13,33,38,150,151,152,153,154] and (B) non-stomatal leaf traits [61,62,92,155,156,157,158,159,160,161,162,163,164,165,166,167,168,169,170,171,172,173,174,175,176,177,178], encompassing key functional areas such as stomatal development and regulation, cuticular wax biosynthesis, leaf morphology, stress response pathways, and metabolic adaptations.

## 6. Enhanced WUE in Rice

The concept map in Figure 3 encapsulates the complex nature of WUE in rice, highlighting the intricate relationships between stomatal and non-stomatal traits. This visual representation underscores the complexity of WUE as a trait and the need for integrated approaches in future research and breeding efforts. Stomatal traits such as density, size, and conductance interact closely with non-stomatal factors like leaf anatomy, photosynthetic efficiency, and metabolic adaptations. The integration and optimisation branch of the map emphasises the importance of considering trade-offs between these traits and leveraging emerging technologies for crop improvement. This holistic view guides future research directions, suggesting that advancements in WUE will likely come from a synergistic approach that combines traditional breeding with cutting-edge technologies like CRISPR gene editing and high-throughput phenotyping. The map also highlights the critical role of environmental interactions, reminding researchers of the need to develop rice varieties that maintain a high WUE across diverse and changing climatic conditions.

Moving forward, the challenge lies in translating this complex understanding into practical breeding strategies and agronomic practices. Interdisciplinary collaboration will be key, bringing together expertise in plant physiology, genetics, agronomy, and data science. By addressing the interconnected aspects of WUE illustrated in this concept map, researchers can develop more resilient and water-efficient rice varieties, contributing significantly to sustainable agriculture in the face of global water scarcity and climate change.

## 7. Conclusions

There is an intricate relationship between stomatal and non-stomatal leaf traits, and both collectively have an impact on WUE in rice. Understanding the complex interactions among these traits is essential for developing rice varieties with an improved WUE, as well as for implementing agronomic practices that optimise water management in rice cultivation. Further investigations will reveal specific beneficial varieties for special agronomic practices. By focusing on both stomatal and non-stomatal leaf traits, researchers can develop climate-resilient rice varieties that ensure food security for future generations. Insights from this review can guide breeding programs to develop new crop varieties with an enhanced WUE, ensuring sustainable production in water-scarce regions. Precision agriculture practices can be optimised based on specific leaf traits linked to efficient water use. The review offers a foundation for future research aimed at elucidating the mechanisms of WUE, facilitating the discovery of novel genetic targets and pathways.

## Figures and Tables

**Figure 1 biology-14-00843-f001:**
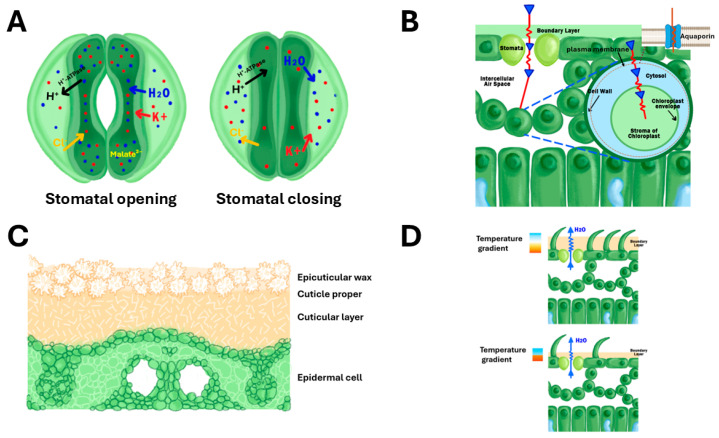
Anatomical structures and physiological mechanisms regulating water use efficiency (WUE) in rice leaves. (**A**) Ion transport mechanisms in guard and subsidiary cells during stomatal dynamics. The left panel illustrates stomatal opening, initiated by activation of plasma membrane H^+^-ATPases, which drive H^+^ efflux and establish a membrane potential that facilitates the influx of K^+^, Cl^−^, and water (H_2_O), resulting in increased guard cell turgor and pore opening. Malate^2−^ is indicated within the guard cell as a key osmotic anion. Right panel shows stomatal closing with efflux of ions, leading to guard cell shrinkage and reduced aperture. (**B**) Carbon dioxide diffusion pathway from atmosphere to chloroplast stroma, showing CO_2_ movement through the boundary layer, plasma membrane, cell wall, intercellular air space, and chloroplast envelope. CO_2_-permeable aquaporins in the plasma membrane facilitate efficient CO_2_ transport, enhancing mesophyll conductance for photosynthesis. (**C**) Cross-sectional view of leaf cuticle architecture showing the multilayer barrier structure: outer epicuticular wax crystals, cuticle proper, cuticular layer, and underlying epidermal cell. This waxy barrier regulates non-stomatal water loss and provides protection against environmental stress. (**D**) Comparison of leaf surface boundary layer effects on temperature gradients and water vapour (H_2_O) diffusion. Upper panel demonstrates how leaf pubescence (trichomes) creates a thicker boundary layer, increasing the temperature gradient and slowing water vapour diffusion, thereby improving water conservation and thermal regulation. Lower panel shows smooth leaf surface with thinner boundary layer and steeper temperature gradient from leaf surface to ambient air.

**Figure 2 biology-14-00843-f002:**
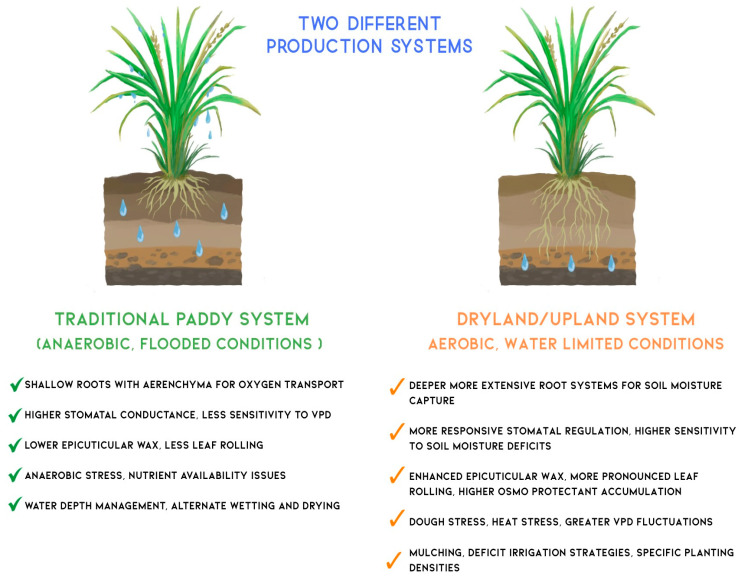
Comparative analysis of physiological adaptations and management strategies in two distinct rice production systems. The diagram contrasts traditional irrigated paddy systems (**left**) operating under anaerobic, flooded conditions with dryland/upland systems (**right**) under aerobic, water-limited conditions. Traditional paddy system features shallow root architecture with aerenchyma tissue for oxygen transport under waterlogged conditions, higher stomatal conductance with reduced sensitivity to vapour pressure deficit (VPD), minimal epicuticular wax deposition, and water depth management through alternate wetting and drying practices. Dryland/upland system exhibits deeper, more extensive root systems for enhanced soil moisture capture, responsive stomatal regulation with a higher degree of sensitivity to soil moisture deficits, increased epicuticular wax accumulation and leaf rolling for water conservation, enhanced osmoprotectant production, and water-conserving management practices, including mulching and deficit irrigation strategies. Water droplets illustrate the different soil moisture distribution patterns characteristic of each system.

**Figure 3 biology-14-00843-f003:**
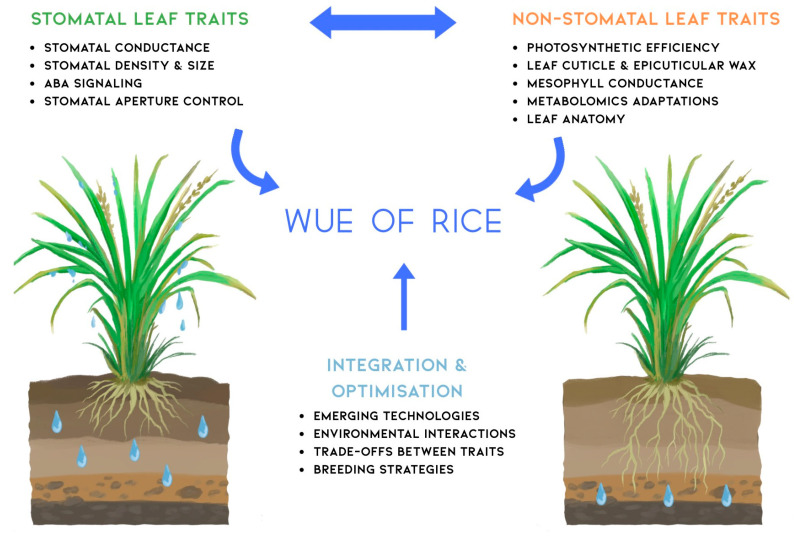
Conceptual framework illustrating the integrated approach to water use efficiency (WUE) improvement in rice through stomatal and non-stomatal leaf traits. The diagram demonstrates how stomatal leaf traits (**left side**: stomatal conductance, density and size, ABA signaling pathways, and aperture control mechanisms) and non-stomatal leaf traits (**right side**: photosynthetic efficiency, leaf cuticle and epicuticular wax deposition, mesophyll conductance, metabolic adaptations, and leaf anatomy) both contribute to overall WUE in rice. The bidirectional arrow between trait categories indicates their interactive and complementary nature in determining water use efficiency. The central integration and optimisation component emphasizes the requirement for holistic approaches incorporating emerging technologies (multi-omics and high-throughput phenotyping), environmental interactions (genotype × environment effects), trade-offs between traits (balancing water conservation with carbon assimilation), and strategic breeding approaches (marker-assisted selection and genomic selection) to develop water-efficient rice cultivars. The rice plants represent the two major cultivation systems (traditional paddy and dryland/upland) discussed throughout the review, highlighting that WUE optimisation strategies must be adapted to specific production environments. This framework provides a roadmap for researchers and breeders to systematically approach WUE improvement through coordinated manipulation of multiple physiological and anatomical traits.

**Table 1 biology-14-00843-t001:** Multi-omics approaches for studying water use efficiency (WUE) in rice. The table summarises key technologies, their specific applications to WUE research, notable research findings, and representative references across different omics platforms. Genomics approaches focus on gene identification and editing; transcriptomics on expression profiling; proteomics on protein-level changes; metabolomics on biochemical responses; phenomics on high-throughput trait measurement; and integrative approaches on system-level understanding of WUE mechanisms.

Omics Level	Key Technologies	Applications to WUE	Notable Findings	Ref
Genomics	GWAS, QTL mapping, CRISPR-Cas9, transgenic *overexpression*	Identification of genomic regions associated with WUE; Targeted gene editing of stomatal regulators	Transgenic overexpression and CRISPR/Cas9-mediated editing of the *OsEPF1* gene in rice have been shown to significantly reduce stomatal density, leading to improved drought tolerance and altered photosynthetic performance. These findings highlight *OsEPF1* as a key regulator of WUE.	[33,91,130]
Transcriptomics	RNA-Seq, microarray analysis, RT-qPCR	Profiling gene expression networks under drought; Comparative transcriptome profiling of drought-tolerant and -sensitive rice genotypes; Identification of drought-responsive noncoding RNAs and their regulatory targets	Transcriptome analysis revealed hundreds of drought-responsive genes, including *OsDREB2A* and *OsLEA3*, which are key regulators in ABA-mediated drought response pathways; 66 miRNAs and 98 lncRNAs were differentially expressed under drought; miR171f-5p targeted Os03g0828701-00, suggesting a role in drought adaptation.	[137,138,139]
Proteomics	LC-MS/MS, iTRAQ, 2D electrophoresis	Identification of drought-responsive proteins; Analysis of PTMs facing water deficit	Over 2000 proteins were detected in rice leaves under drought; 42 showed significant changes. Key drought-responsive proteins included actin depolymerizing factor, S-like ribonuclease, and chloroplastic dehydroascorbate reductase. PTMs such as phosphorylation, ubiquitination, and glycosylation modulate protein function under drought, contributing to stress tolerance.	[139,140]
Metabolomics	GC-MS, LC-MS, NMR spectroscopy	Profiling of osmoprotectants and secondary metabolites under water stress, identifying antioxidant compounds that enhance WUE.	Flag leaves exhibited cultivar-specific increases in proline, sucrose, and malate under combined drought and heat stress. Overaccumulation of flavonoids, such as kaempferol and quercetin, enhances drought and UV tolerance by reducing oxidative damage, overexpressing flavanone 3-hydroxylase showed higher kaempferol and quercetin levels, lower levels of ROS and salicylic acid, and upregulated expression of *DHN* and *UVR8* genes.	[66,141,142]
Phenomics	Thermal imaging, hyperspectral analysis, chlorophyll fluorescence, LiDAR	High-throughput field screening; Non-invasive measurement of physiological traits	Thermal imaging improves detection of crop water deficit by capturing spatial canopy temperature variations, enabling non-invasive and real-time assessments of plant responses to drought stress under field conditions; 3D LiDAR enables precise canopy structure analysis linked to stomatal function and transpiration.	[114,115,124,125]
Integrative Multi-Omics	Network analysis, systems biology, machine learning	Integration of genomic, transcriptomic, proteomic, and metabolomic data; Predictive modelling of WUE traits	Multi-omics integration revealed gene, protein, and metabolite interactions enhancing drought tolerance, highlighting the potential of big omics data to breed drought-resilient rice with improved WUE under climate change. Multi-omics integration provides a holistic view of biological responses to drought stress. Enables identification and manipulation of genes linked to drought tolerance.	[143,144,145]

## Data Availability

Data sharing is not applicable to this article as it is a review of the existing literature. No new data were created or analysed.

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
