# Peer review of "Stomatal and Non-Stomatal Leaf Traits for Enhanced Water Use Efficiency in Rice"

_biology, 2025, doi:10.3390/biology14070843_

Round 1
Reviewer 1 Report (New Reviewer)
Comments and Suggestions for Authors
In this review, the authors examine the relationship between stomatal and non-stomatal leaf traits as they collectively influence WUE in rice. The findings of this review may guide breeding programs to develop new crop varieties with improved WUE. The review is well written and illustrated with figures and tables. However, I have a few comments on the article. (1) I recommend adding a section on the methodology for finding materials for the review, so that it is clear that the authors have collected the most complete picture to date. (2) I think that Table 1 is quite long, it takes up 6 pages, it would be more convenient to place it in the appendix.
• What is the main question addressed by the research? In this review, the authors examine the relationship between stomatal and non-stomatal leaf traits as they collectively influence WUE in rice. The findings of this review may guide breeding programs to develop new crop varieties with improved WUE.
• Do you consider the topic original or relevant to the field? Does it address a specific gap in the field? Please also explain why this is/ is not the case. The findings of this review may guide breeding programs to develop new crop varieties with improved WUE.
• What does it add to the subject area compared with other published material? In recent years, several good reviews have been published on the relationship between stomata and WUE. The authors cite key reviews in the introduction, for example, https://doi.org/10.3389/fpls.2019.00225. The merit of the review I am reviewing is that they have concentrated the information on rice and systematized the key genes.
• What specific improvements should the authors consider regarding the methodology? I recommend adding a section on the methodology for finding materials for the review, so that it is clear that the authors have collected the most complete picture to date.
• Are the conclusions consistent with the evidence and arguments presented and do they address the main question posed? Please also explain why this is/is not the case. At the end, the authors write about the prospects for using the information they have systematized in further research. Undoubtedly, such systematization is useful.
• Are the references appropriate? yes
• Any additional comments on the tables and figures. I think that Table 1 is quite long, it takes up 6 pages, it would be more convenient to place it in the appendix.
Author Response
Please see the attachment.

Reviewer 2 Report (New Reviewer)
Comments and Suggestions for Authors
I enjoyed reading the review of the authors. It is a didactic and thorough review on the analysis of stomata and non-stomata traits and their link with WUE in such an important crop as rice. The topics that follow suggest minor revisions or additional information that could enrich even more the content that has been prepared by the authors.
- The paragraph from lines 58 to 70 seems to be in the middle of other ideas, and rather states the objectives of the review. So its position should be altered.
- In line 74, "water productivity" means that rice is produced in a flooded landscape, right? However, the expression seems a little weird in the context. Please consider improving that.
- Line 137: please italicize the gene symbols such as in this line.
- Line 151: it is a variable not a parameter. Parameter is a population-level measure of any variable.
- Line 159: several genes were listed; their symbols should be italicized when referring to DNA/RNA sequence.
- Lines 124-126: the authors briefly described the variation in stomata density and size and in the field. I think that it is worth adding a whole paragraph explaining the actual findings in the field about stomata variables. That could be at the end of the topic. How do different cultivars react to water scarcity in the field? That should be a guiding question. Is there anything different in the field as to compared with the lab?
- Line 306: instead of “governs”, use “controls”.
- I think that adding information on the heritability of each stomata and non-stomata trait could improve the review, rather than reviewing genes that were overexpressed for specific traits only. That could be added to topic 4.2, guided by the following questions: which traits are more heritable? Which genomics regions have been associated with them, whether with QTL or LD mapping, or both combined? Do the most heritable genes have a strong link with the candidates that have been used for transformation/overexpression? A few genes were already mentioned in that section, but more details should be provided as to how these links were established by QTL or LD mapping.
- Line 438: you can suppress “Modern genetic technique” and rather start with marker-assisted selection.
Author Response
Please see the attachment.

Reviewer 3 Report (New Reviewer)
Comments and Suggestions for Authors
The review paper titled 'Stomatal and Non-Stomatal Leaf Traits for Enhanced Water 2 Use Efficiency in Rice' is an excellent attempt to present comprehensively put together all relevant data of water use efficiency in rice crop.
Introduction is well written and introduces the problem in the field along with the gaps in the current knowledge to address the problem. Review is logically organized to familiarize stomatal and non-stomatal leaf traits followed by integration of both traits in a meaningful way as an attempt to address the problem at hand. Authors have not only cited the relevant literature, but they presented in a easily understood manner with appropriate figures and highly informative tables. This is a very useful resource for someone in the field to quickly find information regarding the gene, Locus ID, its role in general and in relation to WUE. The concept of multi-omics approach for studying water use efficiency in rice is an exemplary one and is highly promising in the current scenario. Future perspectives section is very well thought and is a valuable resource for researchers in the field to help understand the limitations as well as probable approaches to attempt the question.
Author Response
Please see the attachment

This manuscript is a resubmission of an earlier submission. The following is a list of the peer review reports and author responses from that submission.
Round 1
Reviewer 1 Report
Comments and Suggestions for Authors
This manuscript is important considering the significance of the context. It is complex trait and involve interplay of various factors; hence, covering this topic needs comprehensive analyses. The authors carefully tried to add relevant data but the flow and fluency is missing. Following points need to be added for better and clear understanding.
Adding knowledge gaps, contradictions in existing studies, and limitations of current research in section 2 would be beneficial.
Adding comparative insights with other crops or model species could provide broader context to strengthen the review.
In Section 3, the relationship between photosynthesis and WUE needs more clarity. For instance, the sentence in lines 182–183:
“By improving photosynthetic efficiency, plants can achieve higher rates of COâ‚‚ assimilation with minimal water loss, which is particularly beneficial in water-limited environments,”
raises the question how exactly this balance is achieved. A clearer explanation of the physiological mechanism would help readers better understand this connection.
The subheading “ΦPSII determines Photosynthetic efficiency” in subsection 3.1 is also not clear.
Mesophyll conductance is repeated in section 3.2 and 3.5, might be better combined or restructured to avoid repetition and enhance clarity.
Traits like abaxial dominance can influence water loss. This point is missing for WUE improvement.
The abbreviation LMA (Leaf Mass per Area) should be introduced earlier—in line 211, instead of line 269.
Also, in lines 274 and 276, explain the notation “Δ13C”, and check, normally it should have 13 as superscript for scientific correctness. The content in lines 281–287, should also relate to findings in rice.
The subsection 3.8 “Carbon Fixation Efficiency” needs clarification on how it qualifies as a non-stomatal trait, especially since carbon fixation is closely tied to stomatal behavior. A brief explanation distinguishing its role would be helpful.
In Section 5, it would be useful to include advantages and limitations of emerging technologies, as well as a brief comparison of these technologies to traditional methods. This would provide a more balanced view and assist readers in evaluating their practical utility.
Table or Figure demonstrating the linkage of stomatal and non-stomatal traits, description of positive and negative association/correlation with WUE is missing. Moreover, it would be very significant to discuss the interplay of these stomatal and non-stomatal traits, what are the traits that are associated positively and those with vice versa interaction.
WUE, stomatal and non-stomatal traits and their trend under biotic and abiotic stresses would add the value for sustainable agriculture under changing climate
Section of general explanation of stomatal traits and WUE is well developed, as there should be thorough literature review regarding each stomatal trait in relation to WUE and other biotic/ abiotic stresses.
281-286: ‘Erect leaves, which form smaller angles with the stem, better light penetration to lower canopy layers, enhancing overall photosynthetic efficiency, reducing excessive transpiration at the top canopy and potentially improving WUE’ kindly recheck this, how smaller angles allow better penetration of light and then how good light can reduce water losses?
Summary of Multi-Omics approaches table would be better in lieu /additional of list of genes presented here.
Section 4.2: Studies with interplay of mentioned topic/title are missing.
Section 5.3 is not appropriate here; probably better to add in non-stomatal trait portion.
Reviewer 2 Report
Comments and Suggestions for Authors
The review describes the main characteristics of WUE associated with stomatal and non-stomatal traits. This issue is key in relation to rice, given its growing characteristics. This review, along with the literature review, has practical significance, since examples of existing hybrids or transgenic plants are given, as well as information on genes, manipulations with which can contribute to the creation of new varieties with specified characteristics. The review can be published and will be of interest to researchers directly studying rice resistance and optimization of its cultivation.
Reviewer 3 Report
Comments and Suggestions for Authors
Review Stomatal and Non-Stomatal Leaf Traits for Enhanced Water Use Efficiency in Rice by Yvonne Fernando, Mark Adams, Markus Kuhlmann, Vito Butardo Jr is an attempt to generate fragmentary ideas used in the evaluation of various rice genotypes with an emphasis on the need for careful treatment of water sources and mainly with an emphasis on the use of water from limited natural reservoirs in rainfed agriculture.
The manuscript is designed in compliance with the requirements and contains the necessary parts.
I was extremely interested in getting acquainted with the presented materials, but a number of omissions greatly disappointed me. This work should undergo significant revision due to critically dangerous omissions that violate the logic and mislead the authors' recommendations in matters of rice breeding and especially specific genotypes of rainfed rice.
Unfortunately, I cannot recommend this generally good review due to ignoring the most common physiological parameters taken into account by those who work with this crop. I will list the main problems. The authors do not analyze the structure of the aerenchyma in rice plants that allows for transpiration during flooding, what is even worse, they probably believe that respiration is provided by oxygen access possible due to long roots. This incredible assumption and misunderstanding of the processes of root formation in rice is by no means harmless. The structure of roots in the absence of water and flooding conditions are fundamentally different, as is the provision of transpiration in these conditions, which is quite well known.
Even worse, the authors ignore the concept of aerenchyma in leaves. Obviously, in order to get into the plastid, CO2 must first dissolve in water concentrated on the surface of the cell walls - in the intercellular spaces, that is, the larger the surface area, the higher the possibility of evaporation and the easier it is for CO2 to get into the symplast and apoplast and into the cell.
Another problem is the lack of understanding of the options for transpiration and water release in cereals. The authors believe that there are two options: through the stomata and through the epidermis. Is this really so and is there no other system? No, hydathodes are characteristic of grasses! The authors' ideas about the vessels in the leaves are even more surprising.
It is clear from the text that the authors are unfamiliar with the physics of gas solubility and a number of other areas important for understanding the problem; neither omics technologies nor metabolomics will help here. This is just anatomy, just agronomy, just physics. A simple example. The authors mention the role of nitrogen in the processes of adaptation to lack of moisture. However, I could not understand how nitrogen is related to moisture. Their expression and activity, affected by environmental factors like nitrogen supply, impact leaf hydraulic conductance and photosynthetic efficiency [45, 47]. L 200-201 If we assume that we are talking about gas, then it is unclear how a gas that forms the basis of our atmosphere can affect photosynthesis, taking into account its solubility curves; I cannot imagine this. What did the authors mean by "leaf hydraulic conductance"? If this applies to transpiration, then perhaps it should have been formulated differently. There are many such sentences in the review that make sense only if you do not consider them terms. This should be read and clearly written what is meant
Another problem is the use of images that are not related to the object. It is not worth giving a diagram of the stomata of dicots when analyzing the situation with rice. It is not worth using an image of a leaf without normal vascular bundles taking into account companion cells, phloem and xylem cells. Use appropriate images.
In addition, if this review is aimed at formulating problems and identifying key search points, then it should be shown how and what works, where everything is clear, and where there are unresolved issues or where it works in one case one way, and in another differently (which is the case in reality). It is better to draw not very aesthetically, but correctly, than to place color pictures without taking into account the most important parameters.
I recommend providing all sections of anatomy with diagrams, it will immediately become clear what we are talking about.
The physics of the relationship between transpiration and leaf cooling, regulation of overheating and conservation of heat and moisture in different organs must be reflected in detail.
If we are talking about dryland farming and not traditional crops, this should be reflected in the description/introduction and in the diagram.
The dynamics of development at different phases is not reflected in the review at all.
I would also like the authors to take into account that rice forms leaves sequentially and the structure and location of the stomata cannot be changed when the leaf is formed, so it is incorrect to cite works comparing the shape and number of stomata in the upper part of the plant and in the lower part, since these leaves a priori differ in the conditions of their formation, and their structural features and location of the stomata do not reflect the reaction to stress or environmental conditions at the time of the study.
The main problem I find is the lack of setting tasks to identify points that need to be clarified and those correlations that have already been obtained and should be established.
I strongly recommend that this review be corrected.